# The Association between 5-Hydroxytryptamine Receptor 1B rs13212041 Polymorphism and Trait Anxiety in Chinese Han College Subjects

**DOI:** 10.3390/life11090882

**Published:** 2021-08-27

**Authors:** Xiaofei Ruan, Suwen Fang, Qi Zheng, Senqing Qi, Yingfang Tian, Wei Ren

**Affiliations:** 1Key Laboratory of Modern Teaching Technology, Ministry of Education, Xi’an 710062, China; rxf@snnu.edu.cn; 2College of Life Sciences, Shaanxi Normal University, Xi’an 710119, China; 202456@snnu.edu.cn (S.F.); zq182069@snnu.edu.cn (Q.Z.); qisenqing@snnu.edu.cn (S.Q.)

**Keywords:** 5-hydroxytryptamine receptor 1B (5-HT1B), gene polymorphisms, single-nucleotide polymorphisms (SNP), trait anxiety, association

## Abstract

Trait anxiety is a vulnerable personality factor for anxiety and depression. High levels of trait anxiety confer an elevated risk for the development of anxiety and other psychiatric disorders. There is evidence that 5-hydroxytryptamine receptor 1B (5-HT1B) gene polymorphisms play an important role in emotional disorders. Genotyping for four single-nucleotide polymorphisms (SNP) (rs11568817, rs130058, rs6297, and rs13212041) was conducted for 388 high trait anxious (HTA) individuals and 463 low traitanxious (LTA) individuals in Chinese Han college subjects. The results showed that the frequencies of the C-allele and TC + CC genotype of rs13212041 in the LTA individuals were higher than that in the HTA individuals (*p* = 0.025 and *p* = 0.014, respectively). Both the C-allele and TC + CC genotype were associated with trait anxiety decreasing (OR = 0.771 and OR = 0.71, respectively). Furthermore, different gene model analysis also showed that the C allele was a protective factor for trait anxiety in Chinese Han college subjects. These findings suggest that 5-HT1B rs13212014 may play a role in trait anxiety among China Han college subjects. The rs13212014 polymorphism may be involved in decreasing the risk of trait anxiety. These results also provide a novel insight into the molecular mechanism underlying trait anxiety.

## 1. Introduction

Trait anxiety is defined as an individual’s disposition to experience frequent and intense anxiety and worry in response to various stress situations. Individuals with high trait-anxious (HTA) are considered more susceptible to clinical anxiety and depression [1,2], which refers to individuals with HTA as pre-existing forms of anxiety. A large number of behavioral studies have revealed that individuals with HTA exhibit cognitive and emotional disorders, which are similar to anxiety disorders and behavioral biases, such as excessive uneasiness and concern about uncertain events on critical symptoms, threatening information, attention bias, persistent attention, low tolerance for uncertain information, and rejection or dislike of negative results or information [3,4,5]. The reasons for those results may be related to the information processing efficiency caused by anxious emotion [6]. Evidence from these studies indicates that individuals with high trait anxiety have similar behavioral performance compared to anxiety disorders. Most importantly, there is consistency that excessive concerns about uncertainty point to the developmental links between high trait anxiety and anxiety disorders.

Serotonin (5-HT) is an important signaling molecule and neurotransmitter, which is widely distributed in the central nervous system and surrounding tissues. In recent years, molecular genetic studies have shown that dysfunction of the 5-HT is closely related to anxiety, depression, loss of appetite, sleeping nap, decreased activity, sexual dysfunction, endocrine function disorder, etc. 5-HT is involved in the regulation of various mental activities and is closely related to psychiatric diseases. Almost all serotonin receptor subtypes are involved in antidepressant or anxiolytic effects [7]. 5-hydroxytryptamine receptor 1B(5-HT1B), an inhibitory G-protein coupled metabotropic receptor that decreases cAMP, is highly expressed in the striatum, pallidum, accumbens nucleus, substantia nigra, and ventral tegmental area [8]. 5-HT1B plays an important role in regulating serotonergic neurotransmission. It has been reported that the function of 5-HT1Bacts both presynaptically, as an inhibitory autoreceptor located on terminals of serotonin neurons and postsynaptically, as an inhibitory heteroreceptorcontrolling the release of other neurotransmitters [9].

Various animal studies have also demonstrated that 5-HT1B plays a role in anxiety-like and anxiolytic-like effects. 5-HT1B gene knockout mice exhibited reduced anxiety and hyperactivity [10]. Nautiyal and colleagues showed that the forebrain 5-HT1B heteroreceptors expressed during an early postnatal period might contribute to the development of the neural systems underlying adult aggression and proved that distinct heteroreceptors acting during adulthood were involved in mediating impulsivity [11]. Interestingly, mice lacking 5-HT1B autoreceptors presented decreased anxiety in the open field test [9]. Similarly, studies byLin and Parsons [12] indicated that stimulation of 5-HT1B receptor increased anxiety-like behavior in the elevated plus-maze test in rats, suggesting the role of this receptor subtype in the pathology and treatment of anxiety. Non-selective 5-HT1B/1D receptors agonist GR127935 also showed anxiolytic-like properties [13]. The observed antianxiety-like effect might be linked to the postsynaptic 5-HT1B receptors or/and 5-HT1B heteroreceptors [14]. These studies suggested that the 5-HT1B receptor might play an important role in trait anxiety. However, its exact role is yet unclear.

Previous human studies reported the associations between the different polymorphisms in the gene coding for 5-HT1B and alcohol dependence [15,16], alcohol abuse [17], aggressive behaviour [18,19], anger and hostility [20], attention-deficit/hyperactivity disorder (ADHD) [21], substance abuse [22], and schizophrenia [23,24]. Although the current genome-wide literature could not find an association of 5-HT1B with anxiety or related phenotypes, including depression and neuroticism in some European populations [25,26,27], a few studies have reported the associations between the different polymorphisms in 5-HT1B and some phenotypes, including anxiety and depression in Chinese and Americans [28,29,30,31]. The most frequently studied 5-HT1B gene variants are rs6296, rs13212041, rs6297, rs11568817, and rs130058. Both rs11568817 and rs130058 were significantly associated with substance use disorders, while rs11568817 was associated with nicotine dependence in men with ADHD. Alcohol-dependent individuals with rs13212041 CC genotype were more frequent compared to the carriers of the T allele in the group with early onset of alcohol abuse [17]. The 3′-untranslated region (3′-UTR) variant rs13212041 potentially enables the microRNA-mediated regulation of 5-HT1B expression [20]. The variant of rs6296 has been associated with attention-deficit hyperactivity disorder [32], aggressive behavior in children [33], substance misuse disorder, and major depression [31]. These allelic associations with trait anxiety are not consistently found. The etiology and pathogenesis of trait anxiety are still largely unclear. It is commonly believed that trait anxiety results from different underlying neurobiological mechanisms, such as genetic and environmental influences. Thus, we hypothesized that 5-HT1B might be involved in the development of trait anxiety in Chinese Han college students.

To our knowledge, this is the first study to examine the association between 5-HT1B gene polymorphism and trait anxiety in Chinese Han subjects. The results might provide novel insights into the serotonergic regulation mechanisms underlying trait anxiety; it could also help further differentiation of trait anxiety and potential improvement of the prediction for anxiety disorders.

## 2. Materials and Methods

### 2.1. Participants

All participants provided written informed consent. The study protocol was approved by the ethics committee of Shaanxi Normal University. Participants in the study were Chinese Han subjects recruited from freshman or senior years. Subjects were asked to provide venous blood samples and fill out the State-Trait Anxiety Inventory (STAI) (Wenli Li, 1995; Spielberger et al., 1983). The trait anxiety subscale is intended to measure the predisposition to experience chronically high levels of anxiety. The number of participants that filled out State-Trait Anxiety Inventory (STAI) was 2645, with 2529 valid questionnaires being received. Among the 2529 individuals, we filtered the top 25% of the participants as the HTA group (case, 632) and the last 25% of the participants as the LTA group (control, 632), followed by the score of STAI by SPSS quartile method [34,35]. However, some individuals were unwilling to provide blood samples. At last, 851 participants with valid DNA and valid data were enrolled for our study, including 388 individuals with HTA and age- and gender-matched 463 individuals with LTA, as shown in Table 1. The classic case-control research paradigm was conducted in our research. A questionnaire test (post-test) was performed on the subjects before genotyping, according to the self-report of participants. All the participants had no history of long-term medication, no symptoms of mental or neurological disorders.

### 2.2. Blood Collection and DNA Isolation

Peripheral blood samples (2 mL) were obtained from each participant. Genomic DNA was extracted from peripheral blood of cases and controls using the GoldMag-Mini whole blood Genomic DNA Purification Kit (GoldMagCo. Ltd., Xi’an, China), as recommended by the manufacturer’s instructions. DNA concentration was determined by the NanoDrop Lite spectrophotometer (Thermo Fisher Scientific, Waltham, MA, USA). The concentration of all DNA samples was normalized to 20 ng/μL.

### 2.3. SNP Selection

SNP inclusion and screening criteria were as follows: (1) based on the GRCh37 database (http://asia.ensembl.org/Homo_sapiens/Info/Index, accessed on 10 May 2020), we downloaded the ped file and info file for the variations of 5-HT1B (Chromosome 6:77460924-77463491) and nearby regulatory regions (±2 kb)in CHB and CHS population. (2) Using Haploview software, we selected tagSNPs based on HWE > 0.05, MAF > 0.1, Min Genotype > 75%, and Tagger r^2^ > 0.8. (3) as the function of the 5-HT1B gene coding region has been studied extensively, SNPs located within 5-HT1B gene regulatory regions were selected. (4) we selected tagSNPs by combined MassARRAY primer design software, HWE > 0.05, MAF > 0.1 and the call rate > 95% in our study population. (5) based on prior reports, polymorphisms in 5-HT1B have been well studied in other mental disorders, including alcohol abuse [17], ADHD comorbidities [21], anger and hostility [20], schizophrenia [24], anxiety and depression [28,29,30,31], but current knowledge of the association between trait anxiety and 5-HT1B gene polymorphisms in the Chinese Han population that have not been well studied were included. Finally, four polymorphisms in 5-HT1B, including rs11568817, rs130058, rs6297, and rs13212041, were randomly selected in this study.

### 2.4. Genotyping

These four 5-HT1B gene polymorphisms were genotyped according to the procedure of iPLEX single base extension amplification technology. MassARRAY Nano dispenser (Agena Bioscience, San Diego, CA, USA) was used to design primers for the amplification process and single-base extension reactions. SNP genotyping was carried out on the MassARRAYiPLEX (AgenaBiosience, San Diego, CA, USA) platform. Agena Bioscience Typer 4.0 software was used to manage and analyze SNP genotypic data. iPLEX primer for 5-HT1B genotyping in this work, as listed in Table 2. The steps involved in the generation of SNP genotypes using the iPLEX chemistry were based on the manufacturer’s protocol, as following: regions targeted by the multiplex assay are amplified by PCR. PCR products are shrimp alkaline phosphatase (SAP) treated to neutralize unincorporated nucleotides. An extension reaction was then performed to extend the PCR fragments by one base into the SNP site. To remove salts from the iPLEX products prior to mass spectrometry, clean resin ion exchange resin was used. The Nanodispenser RS1000 or other compatible dispenserinstrument is used to transfer resin-cleaned iPlex products (analytes) from 384-well plates to SpectroCHIPs. The mass of the resultant extended fragments is then measured using MALDI-TOF, resulting in a spectrum of distinct mass peaks for the multiplex reaction. In addition, this study also set up double wells for each sample during all the processes (including PCR amplification and mass spectrometry) to ensure the accuracy of the results.

### 2.5. Data Analysis

Quantitative data were shown as median ± standard deviation (SD). The Student’s *t*-test was used to compare the differences of quantitative data, and the χ^2^ test was applied for qualitative data. Deviation from Hardy–Weinberg equilibrium (HWE) of genotypic distribution of each SNP in controls was analyzed using Fisher’s exact test. In addition, Pearson’s χ^2^ and Fisher’s exact tests were used to calculate the allele frequencies of case and control, and MAF in controls was defined as the baseline. After adjusting for age and gender, odds ratios (ORs), and 95% confidence interval (95% CI) were calculated using unconditional logistic regression analysis [36]. The relationship between the selected SNPs and trait anxiety was calculated using genotypic model analysis (codominant, dominant, recessive, over-dominant, and log additive) by SPSS 21 (Chicago, IL, USA) and PLINK software [37]. False-positive report probability (FPRP) analysis was used to evaluate the noteworthy associations of the significant findings. We set 0.2 as an FPRP threshold for an association with genotypes under investigation. Multifactor dimensionality reduction (MDR) (version 3.0.2) was performed to evaluate the SNP–SNP interactions in the risk of trait anxiety. Pairwise linkage disequilibrium (LD) was produced using Haploview 4.2 software. Haplotypes analysis was calculated using unconditional logistic regression analysis.In addition to statistical analysis, we also conducted several bioinformatics analyses for the identified SNPs of 5-HT1B. The SNPs information of 5-HT1B was retrieved from the National Center for Biotechnology Information (NCBI) database of dbSNP (http://www.ncbi.nlm.nih.gov/snp/ (accessed on 10 May 2020)). The bioinformatics tools, including SNPinfo Web Server (https://snpinfo.niehs.nih.gov/snpinfo/index.html (accessed on 3 December 2020)) and HaploReg v4.1 (https://pubs.broadinstitute.org/mammals/haploreg/haploreg.php (accessed on 3 January 2021)) were used to identify the potential functional SNPs in human 5-HT1B Statistical analyses were performed using Microsoft Excel (Microsoft Corporation, Redmond, WA, USA) and SPSS (SPSS 21, Chicago, IL, USA)) statistical package. In the study, all the *p*-value were two-sided, and *p* < 0.05 was defined as statistically significant, whereas a value of corrected *p* < 0.05/4 was considered significant after Bonferroni correction.

## 3. Results

### 3.1. Clinical Characteristics of Samples

The characteristics of the enrolled participants are presented in Table 1. 851 participants were enrolled in this study, including 463 individuals with HTA and 388 individuals with LTA. The average STAI score was higher in the HTA group than in the LTA group (*t* = −71.076, *p* < 0.001). There is no significant difference between the HTA and LTA group in terms of gender (*t* = 1.208, *p* = 0.227) or age (*t* = 0.414, *p* = 0.230).

### 3.2. Basic Characteristic of SNP

Four SNPs in 5-HT1B, including rs11568817, rs130058, rs6297, and rs13212041 (MAF ≥ 0.1), were selected for this study. The basic characteristic of SNPs in the enrolled population is shown in Table 3. All the four SNPs were in HWE in the study (*p* > 0.05). The call rate of all the four SNPs was 100%.

Table 3 shows the basic information and the potential function of the selected SNPs. By HaploReg annotation, we found that the selected SNPs were associated with regulation of promoter and/or enhancer histone marks, DNase, proteins bound, and motifs changed. Based on the SNPinfo web server database, rs13212041 might be related to the binding of hsa-miR-622 and hsa-miR-96. Rs6297 was associated with splicing. Moreover, rs130058 and rs11568817 may be located within a transcription factor binding site (TFBS).

### 3.3. Allele and Genotype Frequencies Analysis of SNPs

A summary of allele and genotype frequencies is presented in Table 4. The allele “C” of rs13212041 in 5-HT1Bwas significantly associated with trait anxiety in the study population (OR = 0.77, 95% CI = 0.61–0.97, *p* = 0.025). Individuals carrying the “TC” genotype were significantly more frequent (OR = 0.70, 95%CI = 0.53–0.94, *p* = 0.019) in the subjects with LTA than those with HTA. However, no significant association was found after Bonferroni correction.

### 3.4. Association Analysis

The association between SNPs genotypes and trait anxiety under various genetic model is shown in Table 5. Our analysis showed that crude analysis of rs13212041 was significantly associated with the developing of trait anxiety under the over-dominant model (OR = 0.72, 95% CI = 0.54–0.96, *p* = 0.024), log-additive model (OR = 0.77, 95% CI = 0.62–0.97, *p* = 0.025), and the dominant model (OR = 0.71, 95% CI = 0.54–0.93, *p* = 0.014). However, no significant association was found after Bonferroni correction.

### 3.5. FPRP and Power Analysis

FPRP analysis was carried out to interrogate whether the significant findings was deserving attention (Table 6). At the prior probability level of 0.1, the significant association for rs13212041 (TC vs. TT, FPRP = 0.189; and T/C + C/C vs. TT, FPRP = 0.146) remained noteworthy.

Rs13212041 polymorphism in 5-HT1B with the risk of trait anxiety were discovered, with power values of 0.641 (TC vs. TT), 0.894 (C vs. T), 0.676 (T/C + C/C vs. TT), 0.700 (T/C vs. T/T + C/C), and 0.889 (log-additive). Especially, the power for the allele and log-additive models was more than 85%, suggesting that the sample size was large enough to discover the differences.

### 3.6. MDR Analysis

MDR was used to analyze the interactions of these four SNPs. The results of the MDR model analysis of the SNP-SNP interactions are demonstrated in Table 7. The results showed that rs13212041 was the best single-locus model to predict trait anxiety (testing accuracy, 0.5455; *p* = 0.005; cross-validation consistency, 10/10). The best multi-loci model was the two-locus model, a combination of rs13212041 and rs6297, with the highest testing accuracy (0.5530) and perfect cross-validation consistency (10/10). As shown in Figure 1, the dendrogram and the Fruchterman–Reingoldgraph described the interactions between these SNPs. The patterns of entropy recapitulate the main and/or interaction effect for each pairwise combination of attributes. The strongest interaction effect was found between rs13212041 and rs6297, with the information gain values of 0.16%, suggesting rs13212041 and rs6297 have synergistic interaction sharing the positive information gain with respect to trait anxiety. Additionally, a combination of rs13212041 and rs6297 was the best model to predict the susceptibility to trait anxiety compared to the single SNP alone.

### 3.7. LD and Haplotypes Analysis

Pairwise linkage disequilibrium (LD) and haplotype analyses were conducted for 5-HT1B variants. Figure 2 revealed a LD block in four SNPs (rs13212041, rs130058, rs6297, and rs11568817) with D’ values > 0.99. The frequencies of haplotypes (ATTT, ATCC, ATCT, CATT, and CTTT) and the result of haplotype analysis is showed in Table 8. The haplotype C_rs13212041_T_rs130058_T_rs6297_A_rs11568817_ may be a protective haplotype in trait anxiety (OR = 0.71, 95% CI = 0.52–0.97, *p* = 0.032).

## 4. Discussion

College students are increasingly reporting common mental health problems, such as depression and anxiety [35]. It is very important to explore the prevalence and genetic causes of trait anxiety of Chinese college students. 5-HT1B receptor plays important roles in multiple behavioral traits, such as locomotion, feeding, and thermoregulation, and also in arterial contractile regulation mechanisms. The 5-HT1B receptor has been the focus of much neuropsychiatric and neuropharmacological research [38]. The intronless human 5-HT1B, located at 6q14.3–q16.3 (GDB 132312), encodes a 390-amino-acid polypeptide. Many polymorphisms in the coding sequence and UTRs were screened, and multiple correlation studies were carried out in 5-HT1B [39]. To the best of our knowledge, this is the first study reporting the association of HTR1B rs13212041 with trait anxiety.

In the present study, we investigated four SNPs in 5-HT1B among 851 individuals of Han Chinese students, including 463 HTA individuals and 388 LTA individuals. According to previously reported observations, there is no functional study available for these SNPs, including rs11568817, rs130058, rs6297, and rs13212041 with trait anxiety, but few studies have investigated their roles in other mental disorders. Evidence of association was found between the functional SNP (rs130058) and alcohol, cocaine, and heroin dependence [40]. The rs130058 SNP within 5-HT1B was demonstrated to have a differential association with increasing suicidal ideation depending on the antidepressant type [41]. Some contribution of the functional promoter combination (rs11568817, rs130058)was found with self-reported anger and hostility among young men [20]. The association between three SNPs (rs11568817, rs130058, rs6297) and the susceptibility to schizophrenia and anxiety disorders has not been previously reported [24]. According to previous studies, only a few studies have investigated the association of rs13212041 polymorphism in 5-HT1B with alcohol dependence [17] and schizophrenia [23]. We demonstrated that rs13212041 in 5-HT1B was significantly associated with the personality development of trait anxiety in the Chinese Han population. The frequency of the TC genotype in LTA individuals was significantly higher than in HTA individuals. Both the C-allele, TC genotype, and TC + CC genotype were significantly associated with LTA. FPRP and statistical power were calculated for the positive findings for the samples [42]. At the prior probability level of 0.1, the significant association for rs13212041 remained noteworthy. Our study provides evidence that rs13212041 may be involved in the protective effect of trait anxiety in China Han college subjects.In future studies, we will explore the potential function of rs13212041 to discover the mechanism of treatment, prevention (through nutritional or environmental changes), and diagnosis.

From the results of bioinformatics analysis, we also found the rs13212041 might affect the proteins bound of 5-HT1B with the FOS gene. FOS genes encode leucine zipper proteins that can dimerise with proteins of the JUN family, thereby forming the transcription factor complex AP-1, which can regulate the expression of 5-HT1B [43]. Based on these, we proposed a hypothesis that rs13212041 could influence the proteins bound of 5-HT1B with the FOS gene, indirectly regulate the downstream gene expression, and affect the expression of 5-HT1B through the gene expression network. The specific mechanism needs to be demonstrated further. Moreover, Jensen et al. characterized the SNP (rs13212041; T1997C) in the distal 3’-UTR of 5-HT1B messenger RNA that disrupts a binding site for miR-96 [44]. 

MicroRNAs are 20–21 nucleotide ribonucleic acids that regulate gene expression by binding to complementary sites on messenger RNA, triggering mRNA degradation and/or inhibition of translation [45,46]. Jensen et al. showed that the rs13212041 polymorphism modulates the gene expression by binding to miR-96, and the C-allele of rs13212041 may attenuate the regulatory function of miR-96 [44]. In our study, the “TC” genotype and “C” allele of 5-HT1B rs13212041 were significantly associated with low trait anxiety. The C-allele of rs13212041 appeared to drive the dominant protective effect of trait anxiety. We presumed the “TC” genotype and “C” allele could disrupt 5-HT1B receptor expression by miR-96. Maybe this pattern suggests that the microRNA-binding site polymorphism has great behavioral effects. Further functional assay is necessary to explore the function and the underlying mechanism of rs13212041 polymorphism.

Given that trait anxiety is a complex disease affected by the interaction of genetic and environmental factors, polygenic or SNP-SNP interaction studies may help to discover the risk factors of trait anxiety. Of note, MDR is a powerful method to detect gene-gene interactions without main gene effects in case-control studies of complex diseases [47]. Further, the MDR was used to analyze the interactions of these four SNPs, the result of the present study suggested that rs13212041 was the best single-locus model to predict trait anxiety and a combination of rs13212041 and rs6297 was the best multi-loci model. The interaction Fruchterman–Reingoldgraph also further confirmed that rs13212041 and rs6297 had a strong correlation, suggesting that HTR1B polymorphisms had an additive effect on the risk of trait anxiety development.

As a fundamental form of genetic variation and inheritance unit, haplotype may affect the phenotypes either directly by affecting promoter activity and protein structure, or indirectly through untyped causal variation near the marker [48]. Therefore, haplotype association is of great significance for revealing the etiology of complex phenotypes. In the study, haplotype analysis implied that haplotype C_rs13212041_T_rs130058_T_rs6297_A_rs11568817_ may be a protective haplotype in trait anxiety. Previously, *HTR1B* haplotypes were associated with *HTR1B* gene expression [20]. These hinted to us that HTR1B haplotypes could be a potential trait anxiety risk factor.

### Limitations

One of the study limitations is a lack of careful determination of trait anxiety phenotypes, using nuclear magnetic resonance (NMR), electroencephalogram (EEG), and other effective anxiety laboratory indicators. In the future, we would like to enlarge sample size and complete the phenotype information to evaluate the association between 5-HT1B SNPs and trait anxiety phenotypes. In addition, although 5-HT1BSNPs might be associated with trait anxiety, the results were not significant after multiple testing correction (*p* < 0.05/4). Thus, the present findings need to be confirmed in future studies with a large sample size. Besides, a replication experiment in a different cohort will strengthen our findings. However, our association analysis firstly displayed that rs13212041 was a susceptible site for anxiety, which might lay a good working foundation for subsequent in-depth experimental molecular research.

## 5. Conclusions

Our study provides a new perspective for understanding the genetic mechanism of trait anxiety personality formation. As far as we know, this is the first study reporting the association of 5-HT1B rs13212041 with trait anxiety. Our results suggest that individuals with rs13212041 C-allele might be associated with the reduced risk of high trait anxiety than carriers with T-allele. Our findings provide novel insights into the serotonergic regulation mechanisms underlying the personality of trait anxiety, which could help in further differentiation of trait anxiety and potential improvement of the therapy for avoiding anxiety disorder.

## Figures and Tables

**Figure 1 life-11-00882-f001:**
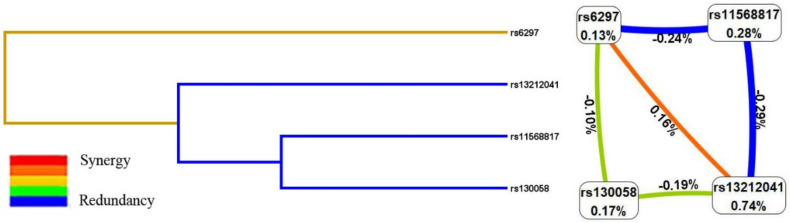
Dendrogram (**left**) and the Fruchterman-Reingold graph (**right**) for the interactions between these SNPs. Positive percent entropy indicates synergy, whereas the negative percent indicates redundancy. The orange line indicated positive interaction, green and blue color indicated weak interactions.

**Figure 2 life-11-00882-f002:**
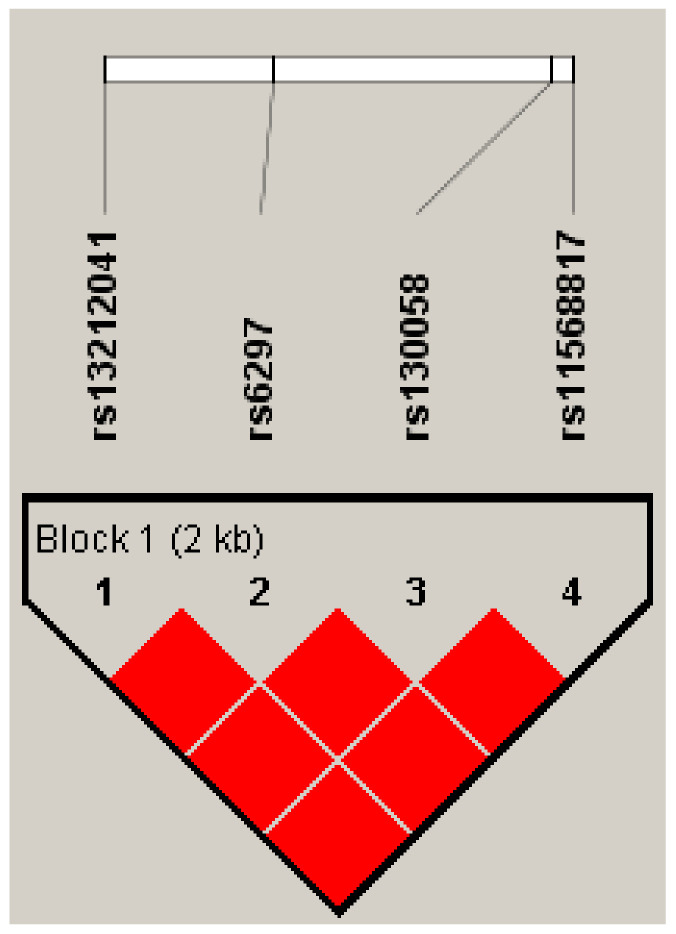
The linkage disequilibrium (LD) plot containing four SNPs from *HTR1B* gene measured by D’.

**Table 1 life-11-00882-t001:** Clinical characteristics.

	LTA	HTA	*t*	*p*-Value
Valid participants	463	388		
Gender			1.208	0.227
Male	86	85		
Female	377	303		
Age (SD)	19.11 ± 2.50	19.04 ± 2.45	0.414	0.230
STAI	31.39 ± 4.11	54.27 ± 5.28	−71.076	<0.001

HTA, high trait anxious; LTA, low trait anxious.

**Table 2 life-11-00882-t002:** TheiPLEX primer list.

SNP_ID	2nd-PCRP	1st-PCRP	UEP_SEQ
rs130058	ACGTTGGATGTCCTCAATTATTCCTCCGCC	ACGTTGGATGTTAGCTAGGCGCTCTGGAAG	GCTGAAACTAGAGGTCA
rs11568817	ACGTTGGATGGTTGTTCCTCTCCACACCG	ACGTTGGATGTTCACCCTCCTGCACTAGAC	cttcaTCTCCACACCGGGTCTTAG
rs13212041	ACGTTGGATGCGATTGTCAAGCCACAACTC	ACGTTGGATGGTAAAGTGACAGGTACATGA	caccCCATTATGTGTGCTAGTGCC
rs6297	ACGTTGGATGGCATTCCATAAACTGATACG	ACGTTGGATGAACTTGGTCCCCAAAGGTCG	agtcAGTGCACAAGTTGACTTGCC

**Table 3 life-11-00882-t003:** Basic characteristic of SNPs in the enrolled population.

rs #	Chr.pos	Region	Minor Allele	*p*-Value for HWE Test		Call Rate	Haploreg	SNPinfo Web Server
				HTA	LTA			
rs13212041	6:77461407	3′-UTR	C	0.356	0.666	100%	Promoter histone marks, DNAse, Proteins bound (FOS)	hsa-miR-622, hsa-miR-96
rs6297	6:77462224	3′-UTR	C	1.000	0.713	100%	Promoter histone marks, Enhancer histone marks, DNAse, Motifs changed (Zbtb3)	Splicing (ESE or ESS)
rs130058	6:77463564	5′-UTR	A	0.781	0.757	100%	Promoter histone marks, Enhancer histone marks, DNAse, Motifs changed	TFBS
rs11568817	6:77463665	5′-UTR	C	0.501	0.633	100%	Promoter histone marks, Enhancer histone marks, DNAse	TFBS

MAF, minor allele frequency; HWE, Hardy–Weinberg equilibrium; HTA, high trait anxious group; LTA, low trait anxious group; TFBS, transcription factor binding sites. #, SNP ID.

**Table 4 life-11-00882-t004:** Association between SNPs and trait anxiety.

SNP	Control *n* (%)	Case *n* (%)	ORs (95% CI)	*p*-Value
rs13212041				
Genotype			-	
TT	255 (55.1)	246 (63.4)	1	-
TC	180 (38.9)	122 (31.4)	0.70 (0.53–0.94)	0.019 *
CC	28 (6.1)	20 (5.2)	0.74 (0.41–1.35)	0.366
Allele				
T	690 (74.5)	614 (79.1)	1	-
C	236 (25.5)	162 (20.9)	0.77 (0.61–0.97)	0.025 *
rs6297				
Genotype			-	
TT	357 (77.1)	306 (78.9)	1	-
TC	98 (21.2)	77 (19.9)	0.92 (0.66–1.28)	0.670
CC	8 (1.7)	5 (1.3)	0.73 (0.24–2.25)	0.780
Allele				
T	812 (87.7)	689 (88.8)	1	-
C	114 (12.3)	87 (11.218)	0.90 (0.67–1.21)	0.484
rs130058				
Genotype			-	
TT	389 (84.0)	315 (81.0)	1	
TA	72 (15.6)	69 (17.8)	0.48 (0.09–2.70)	0.395
AA	2 (0.4)	4 (1.0)	0.41 (0.07–2.23)	0.282
Allele				
T	850 (91.8)	699 (90.1)	1	-
A	76 (8.2)	77 (9.9)	1.23 (0.88–1.72)	0.218
rs11568817				
Genotype			-	
AA	367 (79.3)	295 (76.0)	1	-
AC	92 (19.9)	85 (21.9)	0.46 (0.13–1.59)	0.245
CC	4 (0.9)	8 (2.1)	0.40 (0.12–1.35)	0.127
Allele				
A	826 (89.2)	675 (87.0)	1	-
C	100 (10.8)	101 (13.0)	1.24 (0.92–1.66)	0.158

SNP, single nucleotide polymorphism; CI, confidence interval; OR, odds ratio. Notes: *p*-value wascalculated by logistic regression adjusted for age and gender. * *p* < 0.05. Case, the high trait anxiety group; Control: the low trait anxiety.

**Table 5 life-11-00882-t005:** Association between SNPs genotypes and trait anxiety.

rs #	Group		Adjust Analysis	
	Control (%)	Case (%)	ORs (95%CI)	*p*-Value
rs13212041				
Dominant model				
TT	255 (55.1)	246 (63.4)	1.00	-
T/C + C/C	208 (44.9)	142 (36.6)	0.71 (0.54–0.93)	0.014 *
Recessive model				
T/T + T/C	435 (94.0)	368 (94.8)	1.00	-
C/C	28 (6.0)	20 (5.2)	0.84 (0.47–1.52)	0.057
Overdominant model				
T/T + C/C	283 (61.1)	266 (68.6)	1.00	
T/C	180 (38.9)	122 (31.4)	0.72 (0.54–0.96)	0.024 *
log-Additive model				
0, 1, 2	463 (54.4)	388 (45.6)	0.77 (0.62–0.97)	0.025 *
rs6297				
Dominant model				
TT	357 (77.1)	306 (78.9)	1.00	-
T/C + C/C	106 (22.9)	82 (21.1)	0.90 (0.65–1.25)	0.537
Recessive model				
T/T + T/C	455 (98.3)	383 (98.7)	1.00	
C/C	8 (1.7)	5 (1.3)	0.74 (0.24–2.29)	0.600
Overdominant model				
T/T + C/C	365 (78.8)	311 (80.2)	1.00	-
T/C	98 (21.2)	77 (19.8)	0.92 (0.66–1.29)	0.635
log-Additive model				
0, 1, 2	463 (54.4)	388 (45.6)	0.90 (0.67–1.21)	0.486
rs130058				
Dominant model				
T/T	389 (84.0)	315 (81.2)	1.00	-
A/T + A/A	74 (16.0)	73 (18.8)	1.22 (0.85–1.74)	0.277
Recessive model				
T/T + A/T	461 (99.6)	384 (99.0)	1.00	-
A/A	2 (0.4)	4 (1.0)	2.40 (0.44–13.17)	0.297
Overdominant model				
T/T + A/A	391 (84.4)	319 (82.2)	1.00	-
A/T	72 (15.6)	69 (17.8)	1.17 (0.82–1.69)	0.384
log-Additive model				
0, 1, 2	463 (54.4)	388 (45.6)	1.24 (0.88–1.73)	0.216
rs11568817				
Dominant model				
A/A	367 (79.3)	295 (76.0)	1.00	-
A/C-C/C	96 (20.7)	93 (24.0)	1.21 (0.87–1.67)	0.259
Recessive model				
A/A-A/C	459 (99.1)	380 (97.9)	1.00	-
C/C	4 (0.9)	8 (2.1)	2.42 (0.72–8.08)	0.139
Overdominant model				
A/A-C/C	371 (80.1)	303 (78.1)	1.00	-
A/C	92 (19.9)	85 (21.9)	1.13 (0.81–1.58)	0.466
log-Additive model				
0, 1, 2	463 (54.4)	388 (45.6)	1.24 (0.92–1.66)	0.159

SNP, single nucleotide polymorphism; CI, confidence interval, OR, odds ratio. Notes: * *p* < 0.05 indicates statistical significance. *p*-values were calculated by two-sided χ^2^ tests or Fisher’s exact tests for each genotype distribution by unconditional logistic regression adjusted for age, gender. Case, the HTA group; Control: the LTA group. #, SNP ID.

**Table 6 life-11-00882-t006:** False-positive report of probability values for the associations between 5-HT1B polymorphisms and trait anxiety susceptibility.

Models		OR (95% CI)	*p*	Statistical Power	Prior Probability				
					0.25	0.1	0.01	0.001	0.0001
rs13212041	TC vs. TT	0.70 (0.53–0.94)	0.019	0.641	0.072 *	0.189 *	0.720	0.963	0.996
	C vs. T	0.77 (0.62–0.97)	0.025	0.894	0.079 *	0.206	0.740	0.966	0.997
	T/C + C/C vs. TT	0.71 (0.54–0.93)	0.014	0.676	0.054 *	0.146 *	0.654	0.950	0.995
	T/C vs. T/T + C/C	0.72 (0.54–0.96)	0.024	0.700	0.098 *	0.245	0.781	0.973	0.997
	Log-additive	0.77 (0.62–0.97)	0.025	0.889	0.082 *	0.212	0.747	0.968	0.997

Statistical power was calculated using the number of observations in the subgroup and the OR and *p* values in this table. The level of false-positive report probability threshold was set at 0.2, and noteworthy findings are presented. *—noteworthy findings.

**Table 7 life-11-00882-t007:** SNP–SNP interaction models of 5-HT1B gene analyzed by the MDR method.

Model	Bal. Acc. CV Training	Bal. Acc. CV Testing	CV Consistency	*p*
rs13212041	0.5492	0.5455	10/10	0.0050
rs13212041, rs6297	0.5582	0.5530	10/10	0.0008
rs130058, rs13212041, rs6297	0.5623	0.5480	8/10	0.0003
rs11568817, rs130058, rs13212041, rs6297	0.5658	0.5417	10/10	0.0002

MDR, multifactor dimensionality reduction; Bal. Acc., balanced accuracy; CVC, cross-validation consistency; OR, odds ratio; CI, confidence interval. *p*-values were calculated using χ^2^ tests.

**Table 8 life-11-00882-t008:** Haplotype association with response.

rs13212041	rs6297	rs130058	rs11568817	Freq	OR (95% CI)	*p*-Value
T	T	T	A	0.648	1.00	---
C	C	T	A	0.118	0.87 (0.64–1.18)	0.370
C	T	T	A	0.116	0.71 (0.52–0.97)	0.032 *
T	T	A	C	0.090	1.15 (0.82–1.62)	0.420
T	T	T	C	0.028	1.14 (0.63–2.05)	0.660

* *p* < 0.05 indicates statistical significance.

## Data Availability

The datasets used or analyzed during the current study are available from the corresponding author on reasonable request.

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
