# Peer review of "The Association between 5-Hydroxytryptamine Receptor 1B rs13212041 Polymorphism and Trait Anxiety in Chinese Han College Subjects"

_life, 2021, doi:10.3390/life11090882_

Round 1
Reviewer 1 Report
Ruan et al. report a candidate gene study on trait anxiety in a Chinese Han college student population. Although I read the works of Ruan and colleagues with interest, I have major concerns, which must be addressed before any decision is made on publication in LIFE:
(1) As the authors pointed out in the limitations, authors selected 4 variants to test the significance of the association between the 5-HT1B gene and the trait of interest. However, the results were not significant after addressing the multiple testing issue. Given the multiple testing situation, it is not appropriate that the author highlighted only the single significant variant rs13212041 to declare the association between the candidate gene and the trait anxiety in the study population.
(2) The authors reported the significance of SNP-SNP interaction using a MDR analysis. Although I can see the marginal level of significance the interaction, this showed lack of explanation about biological interaction mechanism between SNPs. If a SNP-SNP interaction is detected between genes, we can interpreted as epistasis. However, the authors did not provide proper biological interpretations of SNP-SNP interaction in this study.
Perhaps, the authors may try to reconstruct haplotypes of the 4 variants and then to test between haplotypes of the 5-HT1B gene and the trait of interest.
Author Response
Response to Reviewer 1 Comments
Point 1: As the authors pointed out in the limitations, authors selected 4 variants to test the significance of the association between the 5-HT1B gene and the trait of interest. However, the results were not significant after addressing the multiple testing issue. Given the multiple testing situation, it is not appropriate that the author highlighted only the single significant variant rs13212041 to declare the association between the candidate gene and the trait anxiety in the study population.
Response 1: We agree with the reviewer. Our finding only represents the relationship between rs13212041 and trait anxiety, not the association between the candidate gene and the trait anxiety. We modified this description in the full text in the revised version (line 19-20, 305-306, 374-381), as following: “These findings suggest that 5-HT1B rs13212014 may play a role in trait anxiety among China Han college subjects”.
Point 2: The authors reported the significance of SNP-SNP interaction using a MDR analysis. Although I can see the marginal level of significance the interaction, this showed lack of explanation about biological interaction mechanism between SNPs. If a SNP-SNP interaction is detected between genes, we can interpreted as epistasis. However, the authors did not provide proper biological interpretations of SNP-SNP interaction in this study.
Perhaps, the authors may try to reconstruct haplotypes of the 4 variants and then to test between haplotypes of the 5-HT1B gene and the trait of interest.
Response 2: We thank the reviewer for this suggestion. Thanks for your suggestion. 1) MDR analysis can be used for gene-gene interaction and intragenic SNP-SNP interaction (PMID: 26924317, PMID: 26587038). In our study, the results of MDR analysis showed that the strongest interaction effect was found between rs13212041 and rs6297, with the information gain values of 0.16%, suggesting rs13212041 and rs6297 have synergistic interaction sharing the positive information gain with respect to trait anxiety. Additionally, a combination of rs13212041 and rs6297 was the best model to predict the susceptibility to trait anxiety compared to the single SNP alone. We added the description in the revised version. We thanks the reviewer for this suggestion. In the follow-up research, we will further explore the SNP function and the biological mechanism of SNP-SNP interaction (line 271-275).
2) As the reviewer suggested, we reconstructed haplotypes of the 4 variants and then performed the LD and haplotypes analysis to test between haplotypes of the 5-HT1B gene and the and trait anxiety in Chinese Han college subjects These results suggest that the haplotype Crs13212041Trs130058Trs6297Ars11568817 may be a protective haplotype in trait anxiety (OR = 0.71, 95% CI = 0.52 - 0.97, P = 0.032). The results were added in the revised version (line 182-183 and 284-290). The discussion was added (line 366-373).

Reviewer 2 Report
The manuscript describes for the first time the association of rs13212041 polymorphism located in the 3’UTR of 5’HT1B gene with trait anxiety. The design of the study and the methods employed are well implemented and the manuscript is clear and well written in general. Some grammatical corrections and typos should be revised. However, the study only evaluates the role of 4 polymorphisms and many more could have been tested in the same iplex experiment without much extra cost, except for the primers. Thus, the impact of the findings is limited but not unimportant.
There are some things that should be addressed, especially in the discussion of the results. Please, find below the most important changes that should be addressed in order to make the text more comprehensible and to improve the discussion.
It will increase the impacto of the manuscript to have data about the error rate of the massarray experiment. The authors included duplicates for each sample, so the numbers regarding error rates per SNP and per sample should be included. Also, for the future if not possible now, a replication experiment in a different cohort will strengthen their findings.
Line 70: describe ADHD
Line 76: I think rs13212014 should be rs13212041
Line 79: specify to which SNP belong these results
Line 80-81: The 3'-untranslated regions (3'-UTR) variant of rs13212041 potential enables the modulation of microRNA-mediated regulation expression of 5-HT1B.
Should be rewritten: The 3'-untranslated region (3'-UTR) variant rs13212041 potentially enables the microRNA-mediated regulation of 5-HT1B expression.
Line 104: If the top and bottom 25% of participants were selected as the HTA and LTA groups respectively, how is it possible that the number of individuals in both groups is so different (463 vs 388)?? 25% of 2529 is 632 individuals, what is different of both numbers. Does it mean that from the 1264 individuals (632*2) only 851 participants had valid DNA? And if so, why? Could not be collected again? Please clarify the selection of participants.
Line 106-108: please add a reference to Table 1 for description of the participants included.
Line 118: microliter symbol should be modified, might be something of the build pdf.
Line 120: e!GRCh37 I asume that the “e!” should not be there
Line 120-122: only the variants within the 5-HT1B where included or a bigger window +- some Mb was included to add variants of nearby regulatory regions?
Line 124-125: please mention which snps were included. I assume this was done manually and not using any software. Some more information will help, were the snps obtained from other papers? From a database? How many snps were included in this step? This could have been done in the first step by downloading the gene +- 1Mb
Line 126-130: point 4), the sentence is not concluded.polymorphisms in. .. that have not been well studied WERE INCLUDED, or selected… Again here I will specify the number of SNPs. Are they all new and were not selected with the previous steps?
Line 130-132: would be interesting to se in the results or somewhere why from all those snps included only 4 have been tested. Was the exclusion criteria their frequency, call rate,, the iplex design that did not fit all in one plex?
Line 141: some minor changes should be done: iPLEX primers for 5-HT1B genotyping in this work are listed in the Supplementary file S1.
Line 146: IVT and resin steps are missing in the description of the iplex method
Line 148: please claruify in the text if the double wells correspond to an experimental replicate (all the process) or only same pcr product run 2 times in the massarray.
Line 183: Table 1: I would rather change all samples by samples included in the study as these number do not refer to all samples mentioned in the methods.
Line 191: bioinformatics analysis? I think this title should not be here
Line 192: Table 2 shows instead of showed
Line 194: please add marks after histone
Line 197: rs are not TFBS, may be located within a TFBS
Line 198: Table 2. Some of this numbers are duplicated in table 3. I will recommend to remove the column “Gene” and (MAF) so there will be more space for the text. Also, there is some wrong formatting.
Line 201: Title is not appropriate are the frequencies are not of genes but of variants, or alleles and genotypes.
Line 204: Please homogenize along the text the number of decimals provided for ORs and CI and P-vals, specially in 3.3 they are shown with 4 decimals.
Line 205: P=0.014, should it be 0.019 as in T3?
Line 217: rs can be removed from the title but “and” is missing before trait.
Line 225: were is not appropriate here, better use was
Line 234: Table 5 title, please write “Polymorphism” without capital letter
Line 242: Use capital letter for P=0.005
Line 245, 322 and figure 1 description: Fruchterman-Reingold please add “graph”
Line 260: should be rewritten as …important to explore…
Line 268: remove “polymorphism” as it is redundant
Line 282: replace had for have
Line 283: Schizophrenia should not be in capital letters
Line 291: stating that the SNP can decrease the risk of HTA in subjects is a very strong sentence. The SNP will not decrease the risk in the population as you can not change the variant that subjects carry. Rather than that, its role may be investigated in order to discover mechanisms to treat it, to prevent it (via nutrition or environmental changes) or to diagnose it.
Line 293-300: the proteins changed are not between both genes as FOS is in a different chromosome. Please re-evaluate the data about its putative functional activity and rewrite. Furthermore, the rs is NOT located in 5’utr and the paragraph is repetitive.
Line 301: it is not a second SNP, it is the same one. All functional discussion should be revised and expanded a little more
Line 318: PMID, should this be included as a normal reference?
Line 323: please add “that” before HTR1B
Line 325: autoreceptors or, space is missing
Line 339: Please describe in the methods and results sections how you performed bonferroni correction and the results obtained.
Line 350: rewrite: “…likely to be protected of the risk of high trait anxiety…carriers of…”
Line 359: remove “Please add:”
Author Response
Response to Reviewer 2 Comments
Point 1: The manuscript describes for the first time the association of rs13212041 polymorphism located in the 3’UTR of 5’HT1B gene with trait anxiety. The design of the study and the methods employed are well implemented and the manuscript is clear and well written in general. Some grammatical corrections and typos should be revised. However, the study only evaluates the role of 4 polymorphisms and many more could have been tested in the same iplex experiment without much extra cost, except for the primers. Thus, the impact of the findings is limited but not unimportant.
There are some things that should be addressed, especially in the discussion of the results. Please, find below the most important changes that should be addressed in order to make the text more comprehensible and to improve the discussion.
It will increase the impact of the manuscript to have data about the error rate of the massarray experiment. The authors included duplicates for each sample, so the numbers regarding error rates per SNP and per sample should be included. Also, for the future if not possible now, a replication experiment in a different cohort will strengthen their findings.
Response 1: We thank the reviewer for this suggestion. 1) The double wells correspond to an experimental replicate (all the process) during all the process (including PCR amplification and mass spectrometry) to ensure the accuracy of the results. The call rate of all the four SNPs were 100% as shown in Table 3 in the revised version (line 209-210).
2) We strongly agree the reviewer. A replication experiment in a different cohort must be do in the future, which will strengthen our findings. The limitations were revised in the revised version (line 390).
Point 2: Line 70: describe ADHD
Response 2: As the reviewer suggested, we added the full name of ADHD as “Attention-deficit/hyperactivity disorder (ADHD)” in the revised version (line 70-71).
Point 3: Line 76: I think rs13212014 should be rs13212041
Response 3: We are sorry for this clerical error. We modified “rs13212014” to “rs13212041” in the revised version (line 77).
Point 4: Line 79: specify to which SNP belong these results
Response 4: We thank the reviewer for this suggestion. We added the SNP (rs13212041) in the revised version (line 80).
Point 5: Line 80-81: The 3'-untranslated regions (3'-UTR) variant of rs13212041 potential enables the modulation of microRNA-mediated regulation expression of 5-HT1B.
Should be rewritten: The 3'-untranslated region (3'-UTR) variant rs13212041 potentially enables the microRNA-mediated regulation of 5-HT1B expression.
Response 5: As the reviewer suggested, we rewritten the sentence in the revised version (line 81-84).
Point 6: Line 104: If the top and bottom 25% of participants were selected as the HTA and LTA groups respectively, how is it possible that the number of individuals in both groups is so different (463 vs 388)?? 25% of 2529 is 632 individuals, what is different of both numbers. Does it mean that from the 1264 individuals (632*2) only 851 participants had valid DNA? And if so, why? Could not be collected again? Please clarify the selection of participants.
Response 6: We are sorry for not describing clearly of the participants. In our study, the number of participants that filled out State-Trait Anxiety Inventory (STAI) was 2645, then 2529 valid questionnaires were received. Among 2529 individuals, we filtered the top 25% of the participants as the HTA group (case, 632) and the last 25% of the participants as the LTA group (control, 632), followed by the score of STAI by SPSS quartile method. However, some individuals were unwilling to provide blood samples. Therefore, 851 participants with valid DNA and valid data were enrolled for our study at last, including 388 individuals with HTA and age- and gender-matched 463 individuals with LTA (Table 1). As you suggested, we modified the description in the revised version (line 106-109).
Point 7: Line 106-108: please add a reference to Table 1 for description of the participants included.
Response 7: As the reviewer suggested. We added a reference to Table 1 in the revised version (line 111).
Point 8: Line 118: microliter symbol should be modified, might be something of the build pdf.
Response 8: We are sorry for this clerical error. We modified the microliter symbol as mL in the revised version (line 126).
Point 9: Line 120: e!GRCh37 I asume that the “e!” should not be there
Response 9: As the reviewer suggested, we deleted the “e!” of “e!GRCh37” in the revised version (line 128).
Point 10: Line 120-122: only the variants within the 5-HT1B where included or a bigger window +- some Mb was included to add variants of nearby regulatory regions?
Response 10: We thank the reviewer for this suggestion. In our study, four selected polymorphisms, including rs11568817, rs130058, rs6297 and rs13212041 were within the 5-HT1B (Chromosome 6:77460924-77463491) and nearby regulatory regions (± 2 kb). We added the description in the revised version (line 130-131).
Point 11: Line 124-125: please mention which snps were included. I assume this was done manually and not using any software. Some more information will help, were the snps obtained from other papers? From a database? How many snps were included in this step? This could have been done in the first step by downloading the gene +- 1Mb
Response 11: We thank the reviewer for this suggestion. Based on the GRCh37 database, 73 SNPs were within5-HT1B (Chromosome 6:77460924-77463491) and nearby regulatory regions (± 2 kb). Subsequently, according to Haploview software (HWE > 0.05, MAF > 0.1, Min Genotype > 75%, and Tagger r2 > 0.8), the function of SNPs, MassARRAY primer design software and our study population (HWE > 0.05, MAF > 0.1 and the call rate > 95%), and based on prior reports, four polymorphisms in 5-HT1B, including rs11568817, rs130058, rs6297 and rs13212041 were randomly selected in this study.
Point 12: Line 126-130: point 4), the sentence is not concluded.polymorphisms in. .. that have not been well studied WERE INCLUDED, or selected… Again here I will specify the number of SNPs. Are they all new and were not selected with the previous steps?
Response 12: We are sorry for not describing clearly of this point. We rewrote the sentence in the revised version (line 134-143) as below:
“4) we selected tagSNPs by combined MassARRAY primer design software, HWE > 0.05, MAF > 0.1 and the call rate > 95% in our study population. 5) based on prior reports, polymorphisms in 5-HT1B have been well studied in other mental disorders including alcohol abuse[17], ADHD comorbidities[21], anger and hostility[20], schizophrenia[24], anxiety and depression[28-31], but current knowledge of the association between trait anxiety and 5-HT1B gene polymorphisms in the Chinese Han population that have not been well studied were included. Finally, four polymorphisms in 5-HT1B, including rs11568817, rs130058, rs6297 and rs13212041 were randomly selected in this study.”
Point 13: Line 130-132: would be interesting to se in the results or somewhere why from all those snps included only 4 have been tested. Was the exclusion criteria their frequency, call rate, the iplex design that did not fit all in one plex?
Response 13: We thank the reviewer for this suggestion.
1) SNP inclusion and screening criteria are based on the GRCh37 database, Haploview software (HWE > 0.05, MAF > 0.1, Min Genotype > 75%, and Tagger r2 > 0.8), the function of SNPs, MassARRAY primer design software and our study population (HWE > 0.05, MAF > 0.1 and the call rate > 95%). Combined prior reports, four polymorphisms in 5-HT1B, including rs11568817, rs130058, rs6297 and rs13212041 were randomly selected in this study.
2) The exclusion criteria was their frequency, call rate, the iplex design did not fit all in one plex.
Point 14: Line 141: some minor changes should be done: iPLEX primers for 5-HT1B genotyping in this work are listed in the Supplementary file S1.
Response 14: As the reviewer suggested, we modified the Supplementary file S1 about iPLEX primers for 5-HT1B genotyping in this work as Table 2 in the revised version (line 152).
Point 15: Line 146: IVT and resin steps are missing in the description of the iplex method
Response 15: We agree with the reviewer comments. We added the IVT and resin steps in the revised version (line 157-160).
Point 16: Line 148: please claruify in the text if the double wells correspond to an experimental replicate (all the process) or only same pcr product run 2 times in the massarray.
Response 16: We thank the reviewer for this suggestion. The double wells correspond to an experimental replicate (all the process) during all the process (including PCR amplification and mass spectrometry). We added this description in the revised version (line 163).
Point 17: Line 183: Table 1: I would rather change all samples by samples included in the study as these number do not refer to all samples mentioned in the methods.
Response 17: As the reviewer suggested, we modified Table 1 in the revised version. In our study, the number of participants that filled out State-Trait Anxiety Inventory (STAI) was 2645, then 2529 valid questionnaires were received. Among 2529 individuals, we filtered the top 25% of the participants as the HTA group (case, 632) and the last 25% of the participants as the LTA group (control, 632), followed by the score of STAI by SPSS quartile method. However, some individuals were unwilling to provide blood samples. Therefore, 851 participants with valid DNA and valid data were enrolled for our study at last, including 388 individuals with HTA and age- and gender-matched 463 individuals with LTA (Table 1).
Point 18: Line 191: bioinformatics analysis? I think this title should not be here
Response 18: We thank the reviewer for this suggestion. We deleted “bioinformatics analysis” in the revised version (line 211).
Point 19: Line 192: Table 2 shows instead of showed
Response 19: We are sorry for this grammatical errors. We modified “showed” to “shows” in the revised version (line 212).
Point 20: Line 194: please add marks after histone
Response 20: As the reviewer suggested, we added marks after histone in the revised version (line 214).
Point 21: Line 197: rs are not TFBS, may be located within a TFBS
Response 21: We thank the reviewer for this suggestion. We modified the description to “Moreover, rs130058 and rs11568817 may be located within a transcription factor binding sites (TFBS).” in the revised version (line 217).
Point 22: Line 198: Table 2. Some of this numbers are duplicated in table 3. I will recommend to remove the column “Gene” and (MAF) so there will be more space for the text. Also, there is some wrong formatting.
Response 22: As the reviewer suggested, we removed the column (Gene and MAF) and modified the formatting in the revised version (Table 3).
Point 23: Line 201: Title is not appropriate are the frequencies are not of genes but of variants, or alleles and genotypes.
Response 23: We thank the reviewer for this suggestion. We modified 3.3 title to “Allele and genotype frequencies analysis of SNPs” in the revised version (line 222).
Point 24: Line 204: Please homogenize along the text the number of decimals provided for ORs and CI and P-vals, specially in 3.3 they are shown with 4 decimals.
Response 24: As the reviewer suggested, we homogenized along the text the number of decimals provided for ORs (2 decimals) and CI (2 decimals) and P-values (3 decimals) in the revised version.
Point 25: Line 205: P=0.014, should it be 0.019 as in T3?
Response 25: We are sorry for this clerical error. We modified “0.014” to “0.019” in the revised version (line 226-227).
Point 26: Line 217: rs can be removed from the title but “and” is missing before trait.
Response 26: As the reviewer suggested, we removed the rs from the title and added “and” in the revised version (line 240).
Point 27: Line 225: were is not appropriate here, better use was
Response 27: We thank the reviewer for this suggestion. We modified “were” to “was” in the revised version (line 248-249).
Point 28: Line 234: Table 5 title, please write “Polymorphism” without capital letter
Response 28: As the reviewer suggested, we modified “Polymorphism” to “polymorphism” in the revised version (line 257).
Point 29: Line 242: Use capital letter for P=0.005
Response 29: We thank the reviewer for this suggestion. We modified “p = 0.005” to “P = 0.005” in the revised version (line 265).
Point 30: Line 245, 322 and: Fruchterman-Reingold please add “graph”
Response 30: As the reviewer suggested, we added “graph” after Fruchterman-Reingold in the revised version (line 268, 363 and Figure 1 description).
Point 31: Line 260: should be rewritten as …important to explore…
Response 31: We thank the reviewer for this suggestion. We modified “important that exploring” to “important to explore” in the revised version (line 297).
Point 32: Line 268: remove “polymorphism” as it is redundant
Response 32: As the reviewer suggested, we removed “polymorphism” in the revised version (line 305).
Point 33: Line 282: replace had for have
Response 33: As the reviewer suggested, we replaced “had” for “have” in the revised version (line 319).
Point 34: Line 283: Schizophrenia should not be in capital letters
Response 34: We thank the reviewer for this suggestion. We modified “Schizophrenia” to “schizophrenia” in the revised version (line 320).
Point 35: Line 291: stating that the SNP can decrease the risk of HTA in subjects is a very strong sentence. The SNP will not decrease the risk in the population as you can not change the variant that subjects carry. Rather than that, its role may be investigated in order to discover mechanisms to treat it, to prevent it (via nutrition or environmental changes) or to diagnose it.
Response 35: We agree with the reviewer. We deleted the sentence as “and can decrease the risk of high trait anxiety” in the revised version (line 328). In future studies, we will explore the potential function of rs13212041 to discover the mechanism of treatment, prevention (through nutritional or environmental changes) or diagnosis. We added the description in the revised version (line 329-331).
Point 36: Line 293-300: the proteins changed are not between both genes as FOS is in a different chromosome. Please re-evaluate the data about its putative functional activity and rewrite. Furthermore, the rs is NOT located in 5’utr and the paragraph is repetitive.
Response 36: We agree with the reviewer. We modified the sentence to “the rs13212041 might affect the proteins bound of 5-HT1B with FOS gene” in the revised version (line 332-333). Besides, we deleted the description of “rs13212041 is located in the 5'-untranslated region of 5-HT1B” in the revised version (line 336).
Point 37: Line 301: it is not a second SNP, it is the same one. All functional discussion should be revised and expanded a little more
Response 37: We thank the reviewer for this suggestion. We deleted the description of “a second SNP”, and modified the functional discussion in the revised version (line 341, 332-345).
Point 38: Line 318: PMID, should this be included as a normal reference?
Response 38: We thank the reviewer for this suggestion. We modified the references (PMID: 15026468 and PMID: 11404819) respectively as [42] and [47] in the revised version (line 325, 359).
Point 39: Line 323: please add “that” before HTR1B
Response 39: As the reviewer suggested, we added “that” before HTR1B in the revised version (line 364).
Point 40: Line 325: autoreceptors or, space is missing
Response 40: We thank the reviewer for this suggestion. Autoreceptorsors is a complete word with no spaces.
Point 41: Line 339: Please describe in the methods and results sections how you performed bonferroni correction and the results obtained.
Response 41: As the reviewer suggested, we added these description of bonferroni correction in the revised version (line 193-194, 227-228 and 239). A value of corrected P < 0.05/4 was considered significant after Bonferroni correction. However, no significant association was found after Bonferroni correction.
Point 42: Line 350: rewrite: “…likely to be protected of the risk of high trait anxiety…carriers of…”
Response 42: As the reviewer suggested, we rewrote the sentence of“…likely to be protected of the risk of high trait anxiety…carriers of…” to “Our results suggest that individuals with rs13212041 C-allele might be associated with the reduced risk of high trait anxiety than carriers with T-allele.” in the revised version (line 399-400).
Point 43: Line 359: remove “Please add:”
Response 43: As the reviewer suggested, we removed “Please add:” in the revised version (line 410).

Round 2
Reviewer 1 Report
The authors addressed my comments with reasonable way, including a more cautious interpretation of association result. Moreover, authors added the haplotype association result which supports original association of 5-HT1B rs13212014 with the trait of interests. However, the L183 should be revised like the following: Haplotype association analysis was conducted using unconditional logistic regression analysis. Otherwise, this manuscript can be accepted in Life journal.